# Sesquiterpenoids and Xanthones from the Kiwifruit-Associated Fungus *Bipolaris* sp. and Their Anti-Pathogenic Microorganism Activity

**DOI:** 10.3390/jof8010009

**Published:** 2021-12-23

**Authors:** Jun-Jie Yu, Ying-Xue Jin, Shan-Shan Huang, Juan He

**Affiliations:** National Demonstration Center for Experimental Ethnopharmacology Education, School of Pharmaceutical Sciences, South-Central University for Nationalities, Wuhan 430074, China; junjieyu98@outlook.com (J.-J.Y.); Yancy6020@163.com (Y.-X.J.); HuangSS1998@126.com (S.-S.H.)

**Keywords:** *Bipolaris* sp., kiwi-associated fungus, sesquiterpenoid, xanthone, anti-pathogenic microorganism activity

## Abstract

Nine previously undescribed sesquiterpenoids, bipolarisorokins A–I (**1**–**9**); two new xanthones, bipolarithones A and B (**10** and **11**); two novel sativene-xanthone adducts, bipolarithones C and D (**12** and **13**); as well as five known compounds (**14**–**1****8**) were characterized from the kiwifruit-associated fungus *Bipolaris* sp. Their structures were elucidated by extensive spectroscopic methods, electronic circular dichroism (ECD), ^13^C NMR calculations, DP4+ probability analyses, and single crystal X-ray diffractions. Many compounds exhibited anti-pathogenic microorganism activity against the bacterium *Pseudomonas syringae* pv. *actinidiae* and four pathogenic microorganisms.

## 1. Introduction

Kiwifruit (*Actinidia chinensis* Planch., Actinidiaceae) is an emerging, healthy, and economical fruit which has become increasingly popular worldwide owing to its flavor and nutritional properties [1]. It is an excellent source of vitamin C and provides balanced nutritional components of minerals, dietary fiber, folate, and health-promoting metabolites [2,3]. China is the leading kiwifruit producer in the world, followed by Italy and New Zealand. The cultivation area and annual output reached 243,000 hm^2^ and 2,500,000 tons at the end of 2020 [4]. Nevertheless, as the cultivation of kiwifruit expands rapidly, many serious diseases such as bacterial canker, soft rot, bacterial blossom blight, brown spot, and root rot are a serious and ongoing threat to kiwifruit production [5,6,7,8,9,10,11,12]. Particularly, the destructive bacterial canker disease, which is associated with an infection by *P*. *syringae* pv. *actinidiae* (Psa), has led to reduced kiwifruit production and huge economic losses worldwide [13,14]. Although the application of copper-based chemicals and streptomycin have played a positive role in the prevention and treatment of bacterial canker, these chemical residues are extremely threatening to human health and the ecological environment [15,16]. Additionally, chemical fungicides easily induce pathogen resistance [17,18]. Thus, it is urgent to develop safer and more effective biological pesticides.

Endophytic microorganisms reside within different tissues of the host plant without causing any disease symptoms and produce various metabolites with different activities [19,20]. Therefore, the endophytic fungi have been proved to be valuable sources of important natural products [21,22]. Some natural products from endophytic fungi play important roles in plant defense systems. Therefore, we carried out the excavation of anti-Psa active substances from metabolites of kiwifruit endophytes and harvested a number of bioactive molecules. For instance, 3-decalinoyltetramic acids and cytochalasins from the kiwifruit endophytic fungus *Zopfiella* sp showed anti-Psa activity [23,24], while imidazole alkaloids ether were characterized as anti-Psa agents from *Fusarium tricinctum* [25]. These discoveries prompted us to search for more novel and bioactive metabolites from kiwifruit-associated fungi. In the current study, a total of eighteen compounds have been isolated from the large-scale fermentation of the kiwifruit-associated fungus *Bipolaris* sp. (Figure 1), which included nine new sativene or longifolene sesquiterpenoids, bipolarisorokins A–I (**1**–**9**); two new xanthones, bipolarithones A and B (**10** and **11**); two novel sativene-xanthone adducts, bipolarithones C and D (**12** and **13**); as well as five known ones (**14**–**1****8**). Their structures were established by means of spectroscopic methods, namely, ECD and ^13^C NMR calculations, DP4+ probability analyses, and single crystal X-ray diffractions. All compounds were evaluated for their inhibitory activities against Psa. Additionally, their inhibitory activity against four phytopathogens (*Phytophthora infestans*, *Alternaria solani*, *Rhizoctonia solani*, and *Fusarium oxysporum*) were assessed. Here, the details of isolation, structural elucidation, and bioactivity evaluations for **1**–**1****8** are reported.

## 2. Materials and Methods

### 2.1. General Experimental Procedures

Melting points were obtained on an X-4 micro melting point apparatus. Optical rotations were measured with an Autopol IV polarimeter (Rudolph, Hackettstown, NJ, USA). UV spectra were obtained using a double beam spectrophotometer UH5300 (Hitachi High-Technologies, Tokyo, Japan). IR spectra were obtained by a Shimadzu IRTracer-100 spectrometer using KBr pellets. 1D and 2D NMR spectra were run on a Bruker Avance III 600 MHz spectrometer with TMS as an internal standard. Chemical shifts (δ) were expressed in ppm with references to the solvent signals. High resolution electrospray ionization mass spectra (HR-ESIMS) were recorded on a LC-MS system consisting of a Q Exactive™ Orbitrap mass spectrometer with an HRESI ion source (ThermoFisher Scientific, Bremen, Germany) used in ultra-high-resolution mode (140,000 at m/z 200) and a UPLC system (Dionex UltiMate 3000 RSLC, ThermoFisher Scientific, Bremen, Germany). Column chromatography (CC) was performed on silica gel (200–300 mesh, Qingdao Marine Chemical Ltd., Qingdao, China), RP-18 gel (20–45 µm, Fuji Silysia Chemical Ltd., Kasugai, Japan), and Sephadex LH-20 (Pharmacia Fine Chemical Co. Ltd., Uppsala, Sweden). Medium-pressure liquid chromatography (MPLC) was performed on a Büchi Sepacore System equipped with a pump manager C-615, pump modules C-605, and a fraction collector C-660 (Büchi Labortechnik AG, Flawil, Switzerland). Preparative high-performance liquid chromatography (prep-HPLC) was performed on an Agilent 1260 liquid chromatography system equipped with Zorbax SB-C18 columns (5 μm, 9.4 mm × 150 mm, or 21.2 mm × 150 mm) and a DAD detector. Chiral resolution was achieved by HPLC equipped with a Daicel AD-H column. Fractions were monitored by TLC (GF 254, Qingdao Haiyang Chemical Co. Ltd. Qingdao, China), and spots were visualized by heating silica gel plates sprayed with 10% H_2_SO_4_ in EtOH.

### 2.2. Fermentation, Extraction, and Isolation

The fungus *Bipolaris* sp. was isolated from fresh and healthy stems of kiwifruit plants (*Actinidia chinensis* Planch., Actinidiaceae), which were collected from the Cangxi county of the Sichuan Province (GPS: N 31°12′, E 105°76′) in July 2018. Each fungus was obtained simultaneously from at least three different healthy tissues. The fungus was identified as one species of the genus *Bipolaris* by observing the morphological characteristics and analysis of the internal transcribed spacer (ITS) regions. A living culture (internal number HFG-20180727-HJ32) has been deposited at the School of Pharmaceutical Sciences, South-Central University for Nationalities, China.

This fungal strain was cultured on a potato dextrose agar (PDA) medium at 24 °C for 10 days. The agar plugs were inoculated in 500 mL Erlenmeyer flasks, each containing 100 mL potato dextrose media. Flask cultures were incubated at 28 °C on a rotary shaker at 160 rpm for two days as the seed culture. Four hundred 500 mL Erlenmeyer flasks, each containing 150 mL potato dextrose broth (PDB), were individually inoculated with 25 mL of seed culture and were incubated at 25 °C on a rotary shaker at 160 rpm for 25 days.

The cultures of *Bipolaris* sp. (60 L) were extracted four times by EtOAc to afford a crude extract (32.0 g) which was subjected to CC over silica gel eluted with a gradient of CHCl_3_-MeOH (a gradient from 1:0 to 0:1) to give six fractions, A–F. Fraction B (13.0 g) was fractionated by MPLC CC over RP-18 eluted with MeOH–H_2_O (from 10:90 to 100:0, *v*/*v*) to give twelve sub-fractions (B_1_–B_12_). Fraction B_3_ (1.2 g) was applied to Sephadex LH-20 eluting with CHCl_3_–MeOH (1:1, *v*/*v*) and was further purified by preparative HPLC with MeCN–H_2_O (19:81, *v*/*v*, 4.0 mL/min) to obtain compounds **9** (18.6 mg, retention time (t_R_) = 40 min), **18** (22.6 mg, t_R_ = 15.8 min), **2** (3.3 mg, t_R_ = 32 min), and **1** (5.4 mg, t_R_ = 36 min). Fraction B_5_ (2.1 g) was separated by CC over silica gel with a gradient elution of the CHCl_3_–MeOH system (50:1→0:1) and was prepared by HPLC with MeCN–H_2_O (12:88, *v*/*v*, 4.0 mL/min) to obtain **3** (4.9 mg, t_R_ = 36 min), **4** (14.4 mg, t_R_ = 46 min), **17** (28.3 mg, t_R_ = 43 min), and **5** (2.1 mg, t_R_ = 40 min). Fraction B_6_ (1.8 g) was purified over Sephadex LH-20 eluted with MeOH to give four subfractions (B_6.1_–B_6.4_). Fraction B_6.2_ (210 mg) was purified using semipreparative HPLC with MeOH-H_2_O (28:72, *v*/*v*, 3.0 mL/min) to afford **8** (8.8 mg, t_R_ = 17.8 min) and **7** (9.6 mg, t_R_ = 21.1 min). Fraction B_6.3_ (170 mg) was purified by preparative HPLC with MeCN–H_2_O (23:77, *v*/*v*, 4 mL/min) to yield **6** (4.3 mg, 26 min). Fraction C (4.3 g) was separated by CC over silica gel with a gradient elution of PE-acetone (50:1→0:1) to afford subfractions C_1_–C_8_. Fraction C_2_ (340 mg) was purified by preparative HPLC with MeCN-H_2_O (55:45, *v*/*v*, 4 mL/min) to give **12** (10.3 mg, t_R_ = 38 min), **13** (3.7 mg, t_R_ = 39 min), **14** (3.1 mg, t_R_ = 36 min) and **15** (3.4 mg, t_R_ = 34 min). Fraction C_5_ (230 mg) was isolated by CC over Sephadex LH-20 (MeOH) and was prepared by HPLC (32:68, *v*/*v*, 4 mL/min) to give **10** (3.7 mg, t_R_ = 28 min), **11** (4.2 mg, t_R_ = 29 min), and **16** (5.1 mg, t_R_ = 24 min).

Bipolarisorokin A (**1**): colorless crystals; mp 145–148 °C; [*α*]^20^_D_ + 67.8 (*c* 0.01, MeOH); UV (MeOH) *λ*_max_ (log *ε*) 205 (3.30); IR (KBr) *ν*_max_ 3360, 2947, 2833, 1651, 1454, 1114, 1031 cm^−1^; ^1^H NMR (600 MHz, CDCl_3_) and ^13^C NMR (150 MHz, CDCl_3_) data, see Table 1; positive ion HRESIMS *m/z* 251.16624 [M–H]^+^, (calculated for C_15_H_23_O_3_^−^ 251.16527).

Bipolarisorokin B (**2**): colorless oil; [*α*]^22^_D_ − 100.1 (*c* 0.05, MeOH); UV (MeOH) *λ*_max_ (log *ε*) 210 (3.23); ^1^H NMR (600 MHz, CDCl_3_) and ^13^C NMR (150 MHz, CDCl_3_) data, see Table 1; positive ion HRESIMS *m/z* 275.16166 [M+Na]^+^, (calculated for C_15_H_24_O_3_Na^+^ 275.16177).

Bipolarisorokin C (**3**): colorless, needle-like crystals (MeOH); mp 135–138 °C; [*α*]^22^_D_ − 21.8 (*c* 0.05, MeOH); UV (MeOH) *λ*_max_ (log *ε*) 265 (3.49) nm; ^1^H NMR (600 MHz, methanol-*d*_4_) and ^13^C NMR (150 MHz, methanol-*d*_4_) data, see Table 1; positive ion HRESIMS *m/z* 253.17971 [M+H]^+^ (calculated for C_15_H_25_O_3_^+^ 253.17982).

*Bipolarisorokin* D (**4**): colorless oil; [*α*]^25^_D_ + 32.0 (*c* 0.05, MeOH); UV (MeOH) *λ*_max_ (log *ε*) 255 (3.65); ^1^H NMR (600 MHz, methanol-*d*_4_) and ^13^C NMR (150 MHz, methanol-*d*_4_) data, see Table 2; positive ion HRESIMS *m/z* 275.16153 [M+Na]^+^ (calculated for C_15_H_24_NaO_3_^+^ 275.16177). *Bipolarisorokin E* (**5**): colorless oil; [*α*]^25^_D_ − 22.7 (*c* 0.05, MeOH); UV (MeOH) *λ*_max_ (log *ε*) 210 (3.24); ^1^H NMR (600 MHz, methanol-*d*_4_) and ^13^C NMR (150 MHz, methanol-*d*_4_) data, see Table 2; positive ion HRESIMS *m/z* 221.15529 [M–H]^−^ (calculated for C_14_H_21_O_2_^−^ 221.15470).

*Bipolarisorokin F* (**6**): white powder; [*α*]^20^_D_ − 3.3 (*c* 0.04, MeOH); UV (MeOH) *λ*_max_ (log *ε*) 215 (3.72); ^1^H NMR (600 MHz, CDCl_3_) and ^13^C NMR (150 MHz, CDCl_3_) data, see Table 2; positive ion HRESIMS *m/z* 225.18506 [M+H]^+^ (calculated for C_14_H_25_O_2_^+^ 225.18491).

*Bipolarisorokin G* (**7**): colorless oil; [*α*]^20^_D_ + 17.2 (*c* 0.02, MeOH); UV (MeOH) *λ*_max_ (log *ε*) 230 (3.21); ^1^H NMR (600 MHz, CDCl_3_) and ^13^C NMR (150 MHz, CDCl_3_) data, see Table 3; positive ion HRESIMS *m/z* 275.20059 [M+H]^+^ (calculated for C_18_H_27_O_2_^+^ 275.20056).

*Bipolarisorokin H* (**8**): colorless oil; [*α*]^25^_D_ − 136.9 (*c* 0.05, MeOH); UV (MeOH) *λ*_max_ (log *ε*) 225 (3.93); ^1^H NMR (600 MHz, CDCl_3_) and ^13^C NMR (150 MHz, CDCl_3_) data, see Table 3; positive ion HRESIMS *m/z* 277.17984 [M+H]^+^ (calculated for C_17_H_25_O_3_^+^ 277.17982).

*Bipolarisorokin I* (**9**): colorless crystals; mp 191–194 °C; [*α*]^22^_D_ + 8.8 (*c* 0.05, MeOH); UV (MeOH) *λ*_max_ (log *ε*) 210 (3.46); ^1^H NMR (600 MHz, methanol-*d*_4_) and ^13^C NMR (150 MHz, methanol-*d*_4_) data, see Table 3; positive ion HRESIMS *m/z* 251.16621 [M–H]^−^, (calculated for C_21_H_23_O_3_^−^ 251.16527).

*Bipolarithone A* (**10**): colorless oil; [*α*]^23^_D_ + 136.0 (*c* 0.05, MeOH); UV (MeOH) *λ*_max_ (log *ε*) 245 (3.30); ^1^H NMR (600 MHz, CDCl_3_) and ^13^C NMR (150 MHz, CDCl_3_) data, see Table 4; positive ion HRESIMS *m/z* 349.09143 [M+H]^+^, (calculated for C_17_H_17_O_8_^+^ 349.09179).

*Bipolarithone B* (**11**): colorless oil; [*α*]^23^_D_ − 24.2 (*c* 0.05, MeOH); UV (MeOH) *λ*_max_ (log *ε*) 245 (3.30); ^1^H NMR (600 MHz, CDCl_3_) and ^13^C NMR (150 MHz, CDCl_3_) data, see Table 4; positive ion HRESIMS *m/z* 349.09157 [M+H]^+^, (calculated for C_17_H_17_O_8_^+^ 349.09179).

*Bipolarithone C* (**12**): colorless oil; [*α*]^25^_D_ + 52.9 (*c* 0.5, MeOH); UV (MeOH) *λ*_max_ (log *ε*) 245 (4.06); ^1^H NMR (600 MHz, CDCl_3_) and ^13^C NMR (150 MHz, CDCl_3_) data, see Table 5; positive ion HRESIMS *m/z* 541.24310 [M+H]^+^, (calculated for C_30_H_37_O_9_^+^ 541.24321).

*Bipolarithone D* (**13**): colorless oil; [*α*]^25^_D_ + 10.2 (*c* 0.5, MeOH); UV (MeOH) *λ*_max_ (log *ε*) 245 (3.88); ^1^H NMR (600 MHz, CDCl_3_) and ^13^C NMR (150 MHz, CDCl_3_) data, see Table 5; positive ion HRESIMS *m/z* 541.24316 [M+H]^+^, (calculated for C_30_H_37_O_9_^+^ 541.24321).

Crystal data for Cu_**1**_0m: C_15_H_24_O_3_, *M* = 252.34, a = 9.7038(6) Å, b = 13.7866(8) Å, c = 16.6333(10) Å, *α* = 95.329(3)°, *β* = 104.898(2)°, *γ* = 102.525(3)°, *V* = 2073.0(2) Å^3^, *T* = 100(2) K, space group P 1, Z = 6, *μ*(Cu K*α*) = 1.54178 mm^−1^, *F*(000) = 828, 82979 reflections measured, 16831 independent reflections (*R_int_* = 0.0695). The final *R_1_* values were 0.0437 (*I* > 2*σ*(*I*)). The final *wR*(*F^2^*) values were 0.1047 (*I* > 2*σ*(*I*)). The final *R_1_* values were 0.0531 (all data). The final *wR*(*F^2^*) values were 0.1143 (all data). The goodness of fit on *F^2^* was 1.039. Flack parameter = −0.10(7). CCDC: 2124305. Available online: https://www.ccdc.cam.ac.uk (accessed on 11 December 2021).

Crystal data for Cu_**3**_0m: C_15_H_24_O_3_, *M* = 252.34, a = 7.0044(5) Å, b = 10.1468(8) Å, c = 20.1433(14) Å, *α* = 90.00°, *β* = 90.00°, *γ* = 90.00°, *V*= 1431.63(18) Å^3^, *T* = 295(2) K, space group P 21 21 21, with Z = 4, *μ*(Cu K*α*) = 1.54178 mm^−1^, *F*(000) = 552, 6263 reflections measured, 2527 independent reflections (*R_int_* = 0.0500). The final *R_1_* values were 0.0519 (*I* > 2*σ*(*I*)). The final *wR*(*F^2^*) values were 0.1538 (*I* > 2*σ*(*I*)). The final *R_1_* values were 0.0719 (all data). The final *wR*(*F^2^*) values were 0.2087 (all data). The goodness of fit on *F^2^* was 1.117. Flack parameter = −0.40(17). CCDC: 2124306. Available online: https://www.ccdc.cam.ac.uk (accessed on 11 December 2021).

Crystal data for Cu_**9**_0m: C_15_H_24_O_3_, *M* = 252.34, a = 6.8634(2) Å, b = 15.0872(4) Å, c = 13.5156(3) Å, *α* = 90.00°, *β* = 90.4010(10)°, *γ* = 90.00°,*V* = 1399.50(6) Å^3^, T = 295(2) K, space group P 1 21 1, with Z = 4, *μ*(Cu K*α*) = 1.54178 mm^−1^, *F*(000) = 552, 32232 reflections measured, 5982 independent reflections (*R_int_* = 0.0279). The final *R_1_* values were 0.0300 (*I* > 2*σ*(*I*)). The final *wR*(*F^2^*) values were 0.0808 (*I* > 2*σ*(*I*)). The final *R_1_* values were 0.0304 (all data). The final *wR*(*F^2^*) values were 0.0812 (all data). The goodness of fit on *F^2^* was 1.057. Flack parameter = −0.01(3). CCDC: 2124307. Available online: https://www.ccdc.cam.ac.uk (accessed on 11 December 2021).

Crystal data for Cu_**17**_0m: C_14_H_24_O_2_, *M* = 224.33, a = 13.6388(2) Å, b = 13.6388(2) Å, c = 13.0174(2) Å, *α* = 90.00°, *β* = 90.00°, *γ* = 90.00°, *V* = 2097.04(7) Å^3^, *T* = 296(2) K, space group P 31 2 1, with Z = 6, *μ*(Cu Kα) = 1.54178 mm^−1^, *F*(000) = 744, 39026 reflections measured, 3033 independent reflections (*R_int_* = 0.0459). The final *R_1_* values were 0.0353 (*I* > 2*σ*(*I*)). The final *wR*(*F^2^*) values were 0.0988 (*I* > 2*σ*(*I*)). The final *R_1_* values were 0.0366 (all data). The final *wR*(*F^2^*) values were 0.1003 (all data). The goodness of fit on *F^2^* was 1.047. Flack parameter =0.01(5). CCDC: 2126101. Available online: https://www.ccdc.cam.ac.uk (accessed on 11 December 2021).

Crystal data for Cu_**18**_0m: C_15_H_26_O_2_, *M* = 238.36, a = 13.1977(2) Å, b = 13.1977(2) Å, c = 8.49040(10) Å, *α* = 90.00°, *β* = 90.00°, *γ* = 90.00°, *V* = 1478.85(5) Å^3^, *T* = 297(2) K, space group P 43, with Z = 4, *μ*(Cu K*α*) = 1.54178 mm^−1^, *F*(000) = 528, 14568 reflections measured, 3063 independent reflections (*R_int_* = 0.0269). The final *R_1_* values were 0.0534 (*I* > 2*σ*(*I*)). The final *wR*(*F^2^*) values were 0.1525 (*I* > 2*σ*(*I*)). The final *R_1_* values were 0.0541 (all data). The final *wR*(*F^2^*) values were 0.1539 (all data). The goodness of fit on *F^2^* was 1.051. Flack parameter =0.12(7). CCDC: 2126105. Available online: https://www.ccdc.cam.ac.uk (accessed on 11 December 2021).

### 2.3. ECD Calculations

The ECD calculations were carried out using the Gaussian 16 software package [26]. Systematic conformational analyses were performed via SYBYL-X 2.1 using the MMFF94 molecular mechanics force field calculation with 10 kcal/mol of cutoff energy [27,28]. The optimization and frequency of conformers were calculated on the B3LYP/6-31G(d) level in the Gaussian 09 program package. The ECD (TDDFT) calculations were performed on the B3LYP/6-311G(d) level of theory with an IEFPCM solvent model (MeOH). The ECD curves were simulated in SpecDis V1.71 using a Gaussian function [29]. The calculated ECD data of all conformers were Boltzmann averaged by Gibbs free energy.

### 2.4. NMR Calculations

All the optimized conformers in an energy window of 5 kcal/mol (with no imaginary frequency) were subjected to gauge-independent atomic orbital (GIAO) calculations of their ^13^C NMR chemical shifts, using density functional theory (DFT) at the mPW1PW91/6-311+G (d,p) level with the PCM model. The calculated NMR data of these conformers were averaged according to the Boltzmann distribution theory and their relative Gibbs free energy. The ^13^C NMR chemical shifts for TMS were also calculated by the same procedures and used as the reference. After the calculation, the experimental and calculated data were evaluated by the improved probability DP4^+^ method [30].

### 2.5. Antibacterial Activity Assay

The bacterium *P*. *syringae* pv. *actinidiae* was donated by Dr. He Yan of Northwest A&F University, China. A sample of each culture was then diluted 1000-fold in fresh Luria-Bertani (LB) (Beijing Solarbio Science & Technology. Co. Ltd., Beijing, China) and incubated with shaking (160 rpm) at 27 °C for 10 h. The resultant mid-log phase cultures were diluted to a concentration of 5 × 10^5^ CFU/mL, then 160 μL was added to each well of the compound-containing plates. Subsequently, 1:1 serial dilutions with sterile PBS of each compound were performed, giving a final compound concentration range from 4 to256 μg/mL. The minimum inhibitory concentration (MIC, with an inhibition rate of ≥90%) was determined by using photometry at OD_600_ nm after 24 h. Streptomycin was used as the positive control.

### 2.6. Anti-Phytopathogens Assay

Four phytopathogens (*Phytophthora infestane*, *Alternaria solani, Rhizoctonia solani*, and *Fusarium oxysporum*) were cultured in PDA with micro glass beads at 27 °C for a week, as well as shaking (160 rpm). Ninety microliters of PDA, together with a 10 μL volume of an aqueous test sample solution, was added into each well of the 96-well plate. The test solutions contained different concentrations of the sample being tested. Then, agar plugs (1 mm^3^) with fresh phytopathogens were inoculated into each well. Subsequently, a two-fold serial dilution in the microplate wells was performed over a concentration range of 4 to 256 μg/mL. Plates were covered and incubated at 27 °C for 24 h. Finally, the minimum inhibitory concentration was determined by observing the plates, with no growth in the well taken as that value. Hygromycin B was used as the positive control.

## 3. Results and Discussion

Bipolarisorokin A (**1**) was isolated as colorless crystals. Its molecular formula of C_15_H_24_O_3_ was determined on the basis of the HR-ESIMS data (measured at *m/z* 251.16624 [M–H]^−^, calculated for C_15_H_23_O_3_^−^ 251.16527), corresponding to four degrees of unsaturation. The ^1^H and ^13^C NMR spectra, in association with the HSQC spectrum, revealed two methyls, four methenes, seven methines, and two quaternary carbons (Table 1). Of them, signals at *δ*_C_ 66.9 (t, C-11), 69.6 (d, C-14), and 74.9 (d, C-15) were identified as the oxygenated methylene and methines. Two olefinic carbons at *δ*_C_ 156.8 (s, C-2) and 103.5 (t, C-12) corresponded to a double bond, which suggested that **1** possessed a tricyclic system. Considering the 15 carbons in **1**, as well as those isolates from the same source, compound **1** was suggested to be a tricyclic sesquiterpenoid. In the ^1^H–^1^H COSY spectrum, a fragment was revealed, as shown with bold lines in Figure 2. The HMBC correlations from to *δ*_H_ 4.94 (H, s, H-12a) and 4.62 (H, s, H-12b), to *δ*_C_ 156.8 (s, C-2), 54.3 (d, C-1) and 43.0 (s, C-3), established the connections between C-12, C-2, and C-1. Further analyses of ^1^H–^1^H COSY, as well as HMBC correlations from *δ*_H_ 0.92 (3H, d, *J* = 6.8 Hz, H-10) to *δ*_C_ 37.6 (d, C-6), 40.5 (d, C-9) and 66.9 (t, C-11), indicated a hydroxy group at C-11. In addition, the connections of C-8/C-3, C-3/C-4, C-3/C-2, and C-3/C-13 were deduced from HMBC correlations from *δ*_H_ 1.05 (3H, s, H-8) to *δ*_C_ 43.0 (s, C-3), 39.9 (t, C-4), 156.8 (s, C-2), and 58.2 (d, C-13). Moreover, the proton of an oxygenated methine at *δ*_H_ 4.02 (H, d, *J* = 5.9 Hz, H-14) showed key correlations to C-13, C-3, and *δ*_C_ 42.2 (d, C-7), which indicated that *δ*_C_ 69.6 (d, C-14) should be placed at C-13. The above 2D NMR data analysis suggested that compound **1** possessed a sativene type sesquiterpene backbone. A ROESY experiment was carried out to establish the relative configuration of **1** (Figure 3). The key correlations of H-13/H-8, H-13/H-6, H-8/H-14, and H-7/H-13 suggested that H-6, H-7, H-8, and H-13 were *β* oriented, while the correlation of H-1/H-9 indicated that H-1 and H-9 were *α*-oriented. Because of the rigid structure and the ROESY correlation of H-8/H-14, both H-14 and H-15 were assigned as an *α* orientation [31]. Finally, the single-crystal X-ray diffraction not only confirmed the planar structure, as elucidated above, but also established the absolute configuration of **1** (Flack parameter = −0.10(7), CCDC: 2124305; Figure 4).

The molecular formula of bipolarisorokin B (**2**) was determined to be C_15_H_24_O_3_ from the HRESIMS data (measured at *m/z* 275.16166 [M+Na]^+^, calculated for C_15_H_24_O_3_Na^+^ 275.16177). Close similarities were observed in the 1D NMR data (Table 1) of compound **1**. However, signals for a methyl (*δ*_H_ 0.94, d, *J* = 6.9 Hz, H-11; *δ*_C_ 16.4, C-11) and an oxygenated quaternary carbon (*δ*_C_ 73.7, C-6) in **2** was suggested to replace the oxymethylene (*δ*_H_ 3.64, overlap, H-11; *δ*_C_ 66.9, C-11) and the methine (*δ*_H_ 1.65, m, H-6; *δ*_C_ 37.6, C-6) in **1**. These observations indicated that the hydroxy group at C-10 in **1** migrated to C-6 in **2**. The observed ^1^H−^1^H COSY cross-peak of H-10 (*δ*_H_ 0.88, 3H, d, *J* = 6.9 Hz) and H-9 (*δ*_H_ 1.57, 1H, m), and H-9/H-11, along with the HMBC correlations from H-10 to C-6, C-9, and C-11 confirmed the above deduction (Figure 2). Furthermore, ROESY correlations of H-13/H-8, H-8/H-14, H-7/H-13, and H-1/H-9 revealed that compounds **2** and **1** shared the same relative configuration. In consideration of its biosynthetic origin, the absolute configuration of compound **2** was identified the same as that of **1**.

Bipolarisorokin C (**3**) was obtained as colorless needles. Its molecular formula of C_15_H_24_O_3_ was determined on the basis of the HR-ESIMS data (measured at *m/z* 253.17971 [M+H]^+^, calculated for C_15_H_25_O_3_^+^ 253.17982), corresponding to four degrees of unsaturation. The ^1^H NMR data (Table 1) showed characteristic signals, including three methyls at *δ*_H_ 0.78 (3H, d, *J* = 6.4 Hz, H-10), 1.06 (3H, d, *J* = 6.4 Hz, H-11), and 2.13 (3H, s, H-12), and the proton of an aldehyde group at *δ*_H_ 10.02 (H, s, H-15). The ^1^H and ^13^C NMR data, in association with the HSQC data, revealed three methyls, four methenes, five methines, and three nonprotonated carbons (Table 1). Preliminary analyses on the 1D NMR data revealed that **3** was likely to be a seco-sativene type sesquiterpenoid. Detailed analyses of the 2D NMR data indicated that the majority of the data of **3** was the same as those of helminthosporol [32], except for a hydroxy group at C-8 (t, *δ*_C_ 64.6) in **3**, which was confirmed by the HMBC correlations from *δ*_H_ 3.63 (H, d, *J* = 11.6 Hz, H-8a) and 3.71 (H, d, *J* = 11.6 Hz, H-8b) to *δ*_C_ 58.0 (s, C-3), 29.6 (t, C-4), 167.0 (s, C-2), and 60.8 (d, C-13) (Figure 2). A ROESY experiment was carried out to establish the relative configuration of **3** (Figure 3). The cross peaks of H-13/H-8a, H-13/H-4b, H-4b/H-6, and H-7/H-14b were observed, which indicated that H-6, H-7, H-8, and H-13 were *β* oriented. Furthermore, single crystal X-ray diffraction established the relative configuration (Flack parameter = −0.40(17), CCDC: 2124306; Figure 4), and the absolute configuration of **3** was determined by ECD calculations, as shown in Figure 5.

Bipolarisorokin D (**4**) was isolated as a colorless oil. The molecular formula was determined to be C_15_H_24_O_3_ according to the HRESIMS spectra (measured at *m/z* 275.16153 [M+Na]^+^, calculated for C_15_H_24_NaO_3_^+^ 275.16177). Compound **4** had the same molecular formula and NMR spectral patterns to that of **3** (Table 2). The key difference was an oxygenated quaternary carbon (*δ*_C_ 73.5, s) in **4** instead of the methine in **3** (*δ*_C_ 46.4, d). The HMBC correlations from H-4a (*δ*_H_ 1.38, m), H-5b (*δ*_H_ 1.61, m), H-7 (*δ*_H_ 3.16, br s), H-10 (*δ*_H_ 1.02, d, *J* = 6.6 Hz), and H-11 (*δ*_H_ 0.80, d, *J* = 6.6 Hz) to *δ*_C_ 73.5 established the quaternary carbon to be C-6. In addition, a methyl (s, *δ*_H_ 1.07, H-8; *δ*_C_ 18.7, C-8) in **4** replaced the oxygenated methylene (*δ*_C_ 64.6) of C-8 in **3**, which was verified by HMBC correlations from H-8 (*δ*_H_ 1.07, s) to C-2 (*δ*_C_ 170.4, s), C-3 (*δ*_C_ 52.0, s), C-4 (*δ*_C_ 32.4, t), and C-13 (*δ*_C_ 55.3, d). Detailed analyses of 2D NMR (HSQC, HMBC, ^1^H-^1^H COSY and ROESY) data confirmed that the other fragments of **4** were the same as those of **3**.

Bipolarisorokin E (**5**) was obtained as a colorless oil. Its molecular formula C_14_H_22_O_2_ was characterized according to HRESIMS (measured at *m/z* 221.15529 [M–H]^-^, calculated for C_14_H_21_O_2_^−^ 221.15470), implying four degrees of unsaturation. The general features of its NMR data closely resembled that of **3** (Table 2). Detailed analyses of 1D and 2D NMR data revealed the differences. At first, the loss of the aldehyde group at C-1 was revealed by the chemical shift of C-1 at *δ*_C_ 124.2, along with the data from ^1^H–^1^H COSY and HMBC spectra as shown in Figure 2. Second, the hydroxy migrated from C-8 to C-12 (*δ*_C_ 59.8, t) as identified by the HMBC correlation from *δ*_H_ 4.06 (2H, m, H-12) to *δ*_C_ 124.2 (d, C-1), 147.2 (s, C-2), and 47.7 (s, C-3). Third, one double bond between C-9 and C-10 was built by HMBC correlations from *δ*_H_ 4.69 (2H, d, *J* = 5.1 Hz, H-10) to *δ*_C_ 22.7 (q, C-11) and 45.2 (d, C-6). The other parts of **5** were elucidated as the same as those of **3** by a detailed analysis of 2D NMR data. 

Bipolarisorokin F (**6**) was purified as white powder, and its molecular formula C_14_H_24_O_2_ was determinded according to HRESIMS (measured at *m/z* 225.18506 [M+H]^+^, calculated for C_14_H_25_O_2_^+^ 225.18491). Analyses of the 1D and 2D NMR data (Table 2) suggested that **6** showed structural similarities to **3.** The distinction between the two compounds was that the *α*,*β*-unsaturated aldehyde group (*δ*_C_ 140.0, C-1; *δ*_C_ 167.0, C-2; *δ*_C_ 190.0, C-15) in **3** was replaced by a carbonyl (*δ*_C_ 212.0, C-1) and a methylene group (*δ*_C_ 50.7, C-2) in **6**. It was supported by HMBC correlations from *δ*_H_ 2.70 (H, br s, H-7), 0.96 (3H, q, *J* = 7.2 Hz, H-12), and 1.72 (H, dd, *J* = 7.9, 5.0 Hz, H-13) to *δ*_C_ 212.0 (s, C-1), 50.7(d, C-2), and the COSY cross-peak of *δ*_H_ 2.10 (1H, m, H-2) and H-12. The hydroxymethyl group (C-8) in **3** was replaced by a methyl group at C-8 (*δ*_C_ 22.1, q) in **6**, as well as the HMBC correlations from *δ*_H_ 1.09 (3H, s, H-8) to C-2, *δ*_C_ 41.8 (s, C-3), *δ*_C_ 36.1 (t, C-4), and *δ*_C_ 54.9 (d, C-13). The key ROESY cross-peak (Figure 3) of H-2/Ha-14 (H, dd, *J* = 10.7, 5.0 Hz, *δ*_H_ 3.85) suggested that H-2 was *β* oriented. Other ROESY data revealed the same patterns to **3**. Finally, regarding the same origin of **6** and **3**, the absolute configuration of **6** was identified to be the same as that of **3,** as depicted.

The molecular formula of bipolarisorokin G (**7**) was assigned as C_18_H_26_O_2_ based on its HRESIMS spectrum (measured at *m/z* 275.20059 [M+H]^+^, calculated for C_18_H_27_O_2_^+^, 275.20056), which contained three more carbon atoms than **3**. The interpretation of the ^1^H and ^13^C NMR data of **7** (Table 3) indicated the same structure skeleton to that of **3**. Analyses of 2D NMR spectra revealed modifications in **7** (Figure 2). HMBC correlations from *δ*_H_ 0.97 (3H, s, H-8) to *δ*_C_ 165.3 (s, C-2), 52.6 (s, C-3), 33.7 (t, C-4), and 63.6 (d, C-13) suggested that a hydroxy group was missing in **7**. In addition, an *α*,*β*-unsaturated ketone group was identified by the HMBC correlations from *δ*_H_ 6.55 (H, dd, *J* = 15.9, 9.6 Hz, H-14), 6.08 (H, d, *J* = 15.9 Hz, H-16), and 2.20 (3H, s, H-18) to *δ*_C_ 198.6 (s, C-17). In the ^1^H−^1^H COSY spectrum, correlations from H-14 to *δ*_H_ 2.22 (H, d, *J* = 9.6 Hz, H-13) and H-16 indicated that the *α*,*β*-unsaturated carboxyl moiety was located at C-13. Finally, the absolute configuration of **7** can be fully resolved by the ECD calculation, as shown in Figure 5.

Bipolarisorokin H (**8**) was obtained as a colorless oil. Its molecular formula, C_17_H_24_O_3_, was inferred from the pseudomolecular ion peak at m/z 277.17984 [M+H]^+^ in the HRESIMS (calculated for C_17_H_25_O_3_^+^ 277.17982). The NMR data of **8** (Table 3) resembled that of **7**, except for the presence of a carboxyl (*δ*_C_ 171.1, C-17) in **8** instead of a carbonyl (*δ*_C_ 198.6, C-17) in **7**, as well as the loss of a methyl group. This was supported by HMBC correlations from *δ*_H_ 6.80 (H, dd, *J* = 15.4, 9.9 Hz, H-14) and 5.81 (H, d, *J* = 15.5 Hz, H-16) to *δ*_C_ 171.1 (s, C-17). Detailed analyses of 2D NMR data suggested that the other data were the same as those of **7**.

Bipolarisorokin I (**9**) was isolated as colorless crystals. Its molecular formula was identified as C_15_H_24_O_3_ by HRESIMS (measured at *m/z* 251.16621 [M–H]^−^, calculated for C_21_H_23_O_3_^−^ 251.16527). All the spectroscopic data indicated similar patterns to those of longifolene [33]. Detailed analyses of 1D and 2D NMR data revealed the differences. Signals at *δ*_C_ 67.0 (d, C-5), 70.5 (d, C-14), and 74.9 (d, C-15) were identified as the oxygenated methines. Therefore, three hydroxyls were suggested to be placed at C-5, C-14, and C-15, respectively, which were identified by the HMBC and ^1^H–^1^H COSY correlations, as shown in Figure 2. Comprehensive analyses of other data suggested that the other parts of **9** were the same as those of longifolene. The relative configuration of **9** was revealed by a ROESY experiment, as shown in Figure 3. The ROESY correlations of Me-10/H-13, H-13/H-5, Me-8/H-13, Me-10/H-9, and Me-10/H-5 indicated these groups were cofacial (assigned as *β* orientation). In addition, the Me-11/H-1 interaction suggested that H-1 should be *α* oriented. Moreover, the coupling constant between H-14 and H-15 (*J*_14,15_ = 6.2 Hz), as well as the ROESY correlations of Me-8/H-14 and Me-8/H-15, suggested that H-14 and H-15 were *α* oriented. Finally, the single-crystal X-ray diffraction not only confirmed the planar structure but also established the absolute configuration of **9** (Flack parameter =0.01(3), CCDC: 2124307; Figure 4).

Bipolarithone A (**10**) was isolated as a yellow oil, and its molecular formula was determined to be C_17_H_16_O_8_ by HRESIMS (measured at *m/z* 349.09143 [M+H]^+^, calculated for C_17_H_17_O_8_^+^ 349.09179). The NMR data (Table 4) of **10** were similar to those of the dechlorinated methyl ester (**16**) isolated in this study [34]. The major difference was that **10** exhibited a dihydrofuran ring rather than a furan ring. HMBC correlations from H-8 (H, d, *J* = 3.9 Hz, *δ*_H_ 5.64) to C-8a (*δ*_C_ 114.7, s), C-7 (*δ*_C_ 170.0, s), C-9 (*δ*_C_ 178.3, s), and C-10a (*δ*_C_ 167.7, s), together with H-5 (H, ddd, *J* = 6.6, 4.4, 3.9 Hz, *δ*_H_ 5.73) to C-10a, C-8a, C-6 (*δ*_C_ 37.7, t), and C-2′ (*δ*_C_ 169.5, s), supported the above assignment. The relative configuration of **10** was identified by the analysis of its ROESY data. The ROESY correlation between H-8 and H-5 indicated that H-8 had the same orientation as H-5 (assigned as an *α* orientation). The calculated ECD of **10** established the configuration of **10,** as shown in Figure 5. Therefore, the structure of **10** was characterized as depicted.

Bipolarithone B (**11**) was isolated as a yellow oil. The HRESIMS spectrum of **11** suggested a molecular formula of C_17_H_16_O_8_ (measured at *m/z* 349.09157 [M+H]^+^, calculated for C_17_H_17_O_8_^+^ 349.09179), the same as that of **10**. The planar structure of **11** was elucidated to be the same as that of **10** by the analysis of its 1D and 2D NMR data. The main difference was suggested as its stereochemistry at C-8 (*δ*_C_ 79.8, d). Analyses of the ^1^H NMR information showed that the coupling constants of H-8, H-5, and H-6 were significantly different from those of **11,** as shown in the Table 4. Furthermore, the ROESY correlation of H-8 (*δ*_H_ 5.63, 1H, d, *J* = 1.7 Hz)/H-5 (*δ*_H_ 5.62, 1H, ddd, *J* = 8.4, 3.8, 1.7 Hz) was not observed in **11**. These data suggested that **11** was an epimer of **10**. The ECD calculation for **11** was performed, and the results of **11** matched well with the experimental ECD curve (Figure 5). Hence, the absolute configuration of **11** can be fully assigned, as shown.

Bipolarithone C (**12**) was assigned a molecular formula of C_30_H_36_O_9_ based on its HRESIMS data (measured at *m/z* 541.24310 [M+H]^+^, calculated for C_30_H_37_O_9_^+^ 541.24321). The NMR data of **12** were very similar to those of bipolenin I (**14**) (Table 5), a novel sesquiterpenoid-xanthone adduct isolated from the fungus *Bipolaris eleusines* [35]. The significant differences were that there was an absence of an aldehyde group and two olefinic carbons, as well as the presence of an additional methine and carbonyl, in **12**. These data suggested that the *α*,*β*-unsaturated aldehyde moiety disappeared in **12**. This assignment was confirmed by the HMBC correlations of *δ*_H_ 2.16 (H, m, H-2), 1.29 (H, m, H-6), 2.56 (1H, br s, H-7), 0.95 (3H, d, *J* = 7.2 Hz, C-12), and 1.90 (H, m, H-13) to *δ*_C_ 50.6 (d, C-2) and 221.6 (s, C-1). The ROESY spectrum displayed similar patterns to those of **14**. Furthermore, a cross peak between H-2 and H-14a (*δ*_H_ 4.05, 1H, dd, *J* = 11.3, 5.1 Hz) confirmed the relative configuration of C-2, as shown. The absolute configuration of **12** was elucidated by the quantum chemistry calculations. At first, the ECD calculations were conducted on the four possible conformers (**12**a–d), using time-dependent density functional theory (TDDFT) at the B3LYP/6-311G (d) level in methanol with the PCM model. The overall calculated ECD spectrum of each configuration was then generated according to the Boltzmann weighting of the conformers. As a result, the calculated ECD curves of **12**a and **12**d matched well with the experimental data (Figure 5). To determine its final structure, the theoretical NMR calculations and DP4+ probabilities were employed. The ^13^C NMR chemical shifts of **12**a and **12**d were calculated at the mPW1PW91/6-311+G (d,p) level in the gas phase. According to the DP4+ probability analyses, **12**a was assigned with 100% probability (see data in the Appendix A). Structurally, compound **12** comprised of a seco-sativene sesquiterpenoid unit and a xanthone unit, whose absolute configurations were in accord with compound **6** and compound **10**, respectively. Therefore, the structure of **12** was established as depicted.

Bipolarithone D (**13**) had the same molecular formula (C_30_H_36_O_9_) as that of **12**, according to their HRESIMS spectra (measured at *m/z* 541.24316 [M + H]^+^, calculated for C_30_H_37_O_9_^+^ 541.24321). The NMR resonances for **13** (Table 5) resembled those of **12**, except that the resonances of C-6′ (*Δ**δ*_C_ + 1.5), H-6′a (*Δ**δ*_H_ + 0.08), and H-6′b (*Δ**δ*_H_ + 0.15) were shifted downfield, while the data H-5′ (*Δ**δ*_H_ − 0.08) were shifted upfield. A detailed comparison of the 1D and 2D NMR data of **13** with that of **12** indicated that the two compounds possessed the same planar structure. The main difference was the stereochemistry at C-8′. A key ROESY correlation of H-5′/H-8′ could be detected in **12** but not in **13.** In addition, the coupling constants of H-8′ in **13** (*J* = 1.8 Hz) were different from that in **12** (*J* = 3.9 Hz). All the data suggested that compound **13** was a C-8′ epimer of **12**. Finally, the absolute configuration of **13** was confirmed by ECD calculations (Figure 5).

Five known compounds were determined as bipolenins I and J (**14** and **15**), dechlorinated methyl ester (**16**), drechslerines A (**17**), and (+)-secolongifolene diol (**18**) by the comparison of their spectral data with that reported in the literature [32,34,35]. In this study, the absolute configurations of compounds **17** and **18** were confirmed by single crystal X-ray diffractions (Figure 6), which could support the absolute configurations of **1**–**9**, **12,** and **13** as depicted in the text, since they were obtained from the same source.

All compounds (**1**–**18**) were evaluated for their anti-Psa activity. As a result, compounds **10** and **15** showed significant inhibitory activity, with MICs of 64 and 16 μg/mL, respectively, while compounds **7**, **11**, **13**, and **16** showed moderate activity, with MICs of 128 μg/mL (Table 6).

In addition, our previous study on chemicals from *B. eleusines* suggested that sativene-xanthone adducts have promising inhibitory activity against plant pathogenic microorganisms [35]. Therefore, all compounds were evaluated for their inhibitory activity against four plant pathogenic microorganisms, including *P*. *infestane*, *A*. *solani*, *R*. *solani*, and *F*. *oxysporum*. As a result, many compounds showed certain inhibitory activity, as given in Table 6.

A brief structure–activity relationship analysis suggested that the aldehyde-containing sativene sesquiterpenoids were more active than the others, while the xanthones or their derivatives showed better inhibitory activities than sativene sesquiterpenoids.

## 4. Conclusions

A total of 18 compounds, including 13 new ones, were characterized from the kiwifruit-associated fungus *Bipolaris* sp. Their structures, with absolute configurations, were established by means of spectroscopic methods. Many compounds possessed anti-Psa activity and inhibitory activity against plant pathogens. It is concluded that *Bipolaris* sp. is rich in sativene sesquiterpenoids and xanthones, and both sativene sesquiterpenoids and xanthones possess potential antimicrobial application prospects. This study also suggested that it is an effective way to find natural anti-Psa agents from kiwifruit-associated fungi.

## Figures and Tables

**Figure 1 jof-08-00009-f001:**
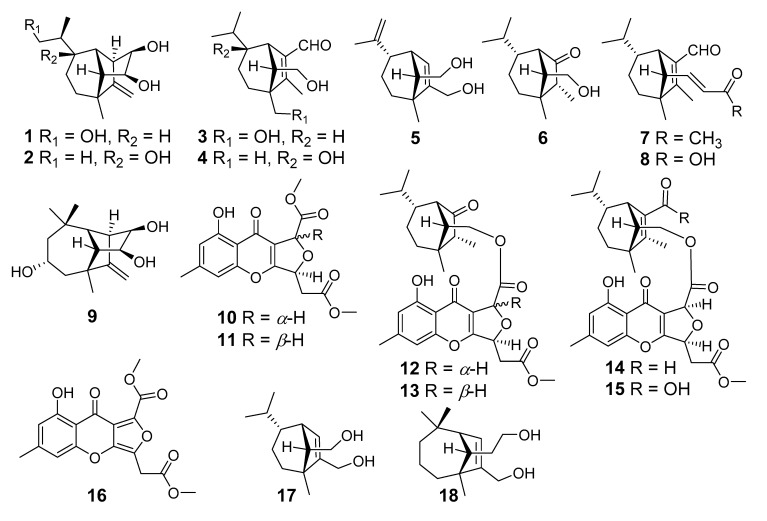
Structures of compounds **1**–**18**.

**Figure 2 jof-08-00009-f002:**
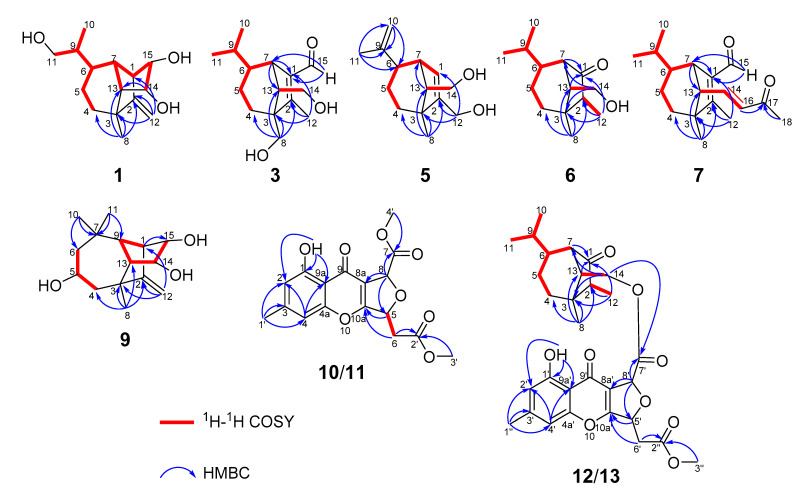
Key ^1^H–^1^H COSY and HMBC correlations for **1**, **3**, **5**, **6**, **7**, and **9**–**13**.

**Figure 3 jof-08-00009-f003:**
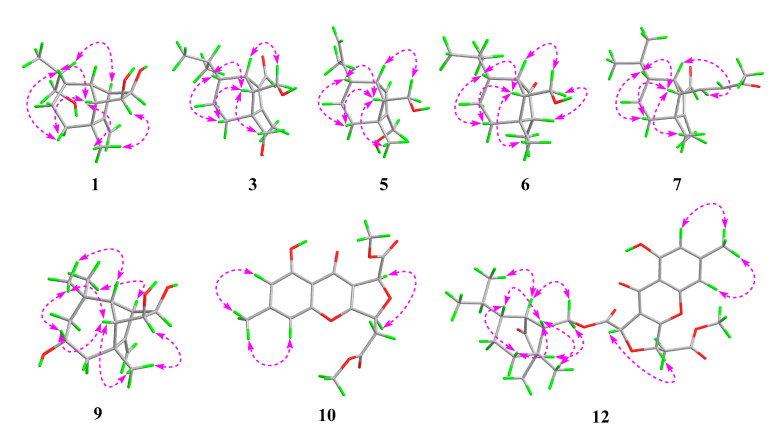
Key ROESY correlations for **1**, **3**, **5**, **6**, **7**, **9**, **10,** and **12**.

**Figure 4 jof-08-00009-f004:**
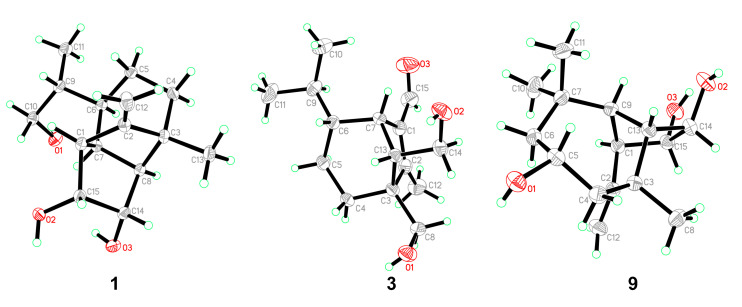
ORTEP diagrams of **1**, **3,** and **9**.

**Figure 5 jof-08-00009-f005:**
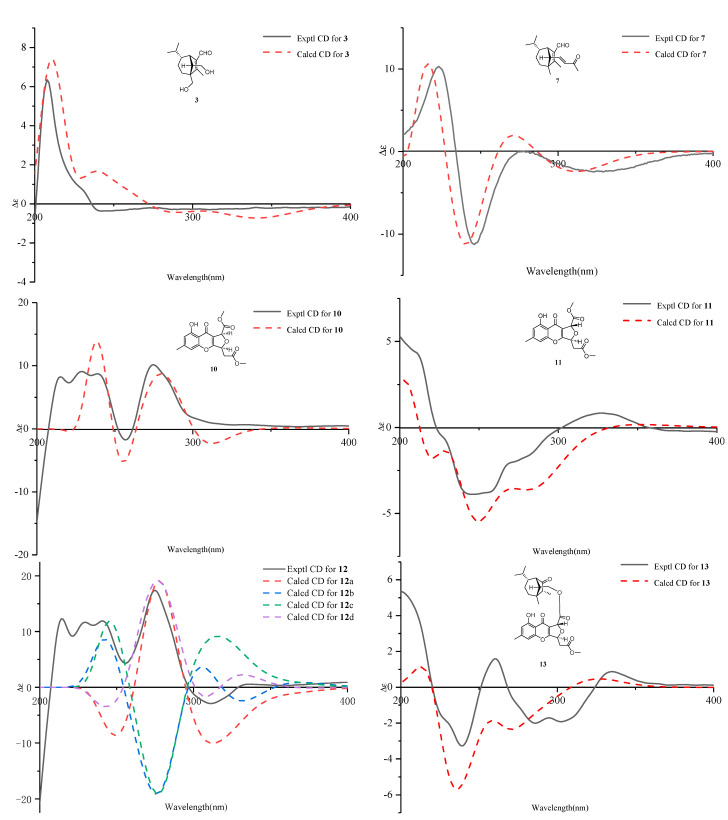
ECD calculations for **3**, **7,** and **10**–**13**.

**Figure 6 jof-08-00009-f006:**
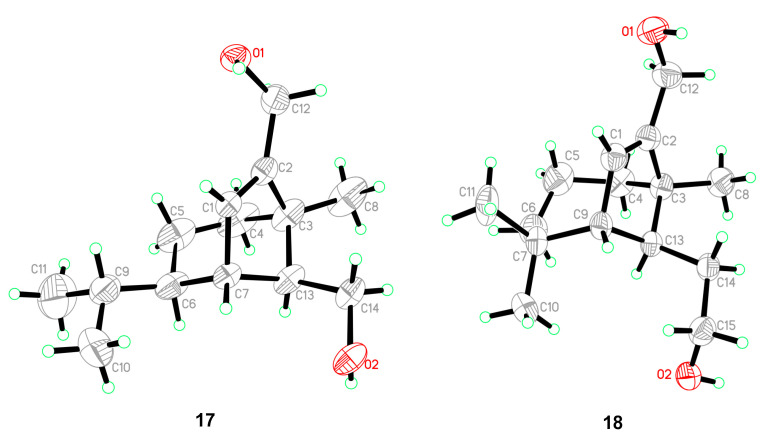
ORTEP diagrams of **17** and **18**.

**Table 1 jof-08-00009-t001:** ^1^H (600 MHz) and ^13^C (150 MHz) NMR Spectroscopic Data for **1**–**3**.

No.	1 *^a^*		2 *^a^*		3 *^b^*	
	*δ*_C_, Type	*δ*_H_ (*J* in Hz)	*δ*_C_, Type	*δ*_H_ (*J* in Hz)	*δ*_C_, Type	*δ*_H_ (*J* in Hz)
1	54.3, CH	2.71, s	55.4, CH	2.59, br s	140,0, C	
2	156.8, C		155.7, C		167.0, C	
3	43.0, C		42.8, C		58.0, C	
4a	39.9, CH_2_	1.50, m	36.7, CH_2_	1.23, m	29.6, CH_2_	1.42, dd (13.2, 6.0)
4b		1.36, m		1.74, m		1.55, dd (12.8, 6.0)
5a	25.2, CH_2_	1.58, m	32.3, CH_2_	1.44, m	25.9, CH_2_	0.90, m
5b		1.24, m		1.55, m		1.80, m
6	37.6, CH	1.65, m	73.7, C		46.4, CH	1.05, m
7	42.2, CH	2.46, s	47.8, CH	2.44, br s	42.7, CH	3.06, br s
8a	20.8, CH_3_	1.05, s	20.8, CH_3_	1.06, s	64.6, CH_2_	3.63, d (11.6)
8b						3.71, d (11.6)
9	40.5, CH	1.46, m	36.9, CH	1.57, m	32.9, CH	1.02, m
10	15.4, CH_3_	0.92, d (6.8)	16.2, CH_3_	0.88, d (6.9)	21.1, CH_3_	0.78, d (6.4)
11	66.9, CH_2_	3.64, overlap	16.4, CH_3_	0.94, d (6.9)	22.1, CH_3_	1.06, d (6.4)
12a	103.5, CH_2_	4.94, s	105, CH_2_	4.69, s	11.0, CH_3_	2.13, s
12b		4.62, s		4.97, s		
13	58.2, CH	1.70, br s	54.7, CH	1.97, br s	60.8, CH	1.82, m
14a	69.6, CH	4.02, d (5.9)	69.5, CH	4.07, d (6.1)	62.9, CH_2_	3.34, dd (11.0, 6.8)
14b						3.61, dd (11.2, 6.8)
15	74.9, CH	3.65, overlap	74.8, CH	3.68, d (6.1)	190.0, CH	10.02, s

*^a^* Measured in CDCl_3_; *^b^* Measured in methanol-*d*_4_.

**Table 2 jof-08-00009-t002:** ^1^H (600 MHz) and ^13^C (150 MHz) NMR Spectroscopic Data for **4**–**6**.

No.	4 *^b^*		5 *^b^*		6 *^a^*	
	*δ*_C_, Type	*δ*_H_ (*J* in Hz)	*δ*_C_, Type	*δ*_H_ (*J* in Hz)	*δ*_C_, Type	*δ*_H_ (*J* in Hz)
1	140.5, C		124.2, CH	5.56, br s	212.0, C	
2	170.4, C		147.2, C		50.7, CH	2.10, m
3	52.0, C		47.7, C		41.8, C	
4a	32.4, CH_2_	1.38, m	35.2, CH_2_	1.34, m	36.1, CH_2_	1.44, m
4b		1.71, m		1.41, dd (12.5, 5.2)		1.66, dd (13.7,5.7)
5a	32.8, CH_2_	1.25, m	26.0, CH_2_	1.56, m	26.0, CH_2_	1.80, m
5b		1.61, m				0.87, m
6	73.5, C		45.2, CH	2.03, m	50.1, CH	1.33, m
7	47.9, CH	3.16, br s	45.3, CH	2.74, br s	51.3, CH	2.70, brs
8	18.7, CH_3_	1.07, s	18.9, CH_3_	0.99, s	22.1, CH_3_	1.09, s
9	37.2, CH	1.28, m	150.3, C		29.9, CH	1.55, m
10	17.1, CH_3_	1.02, d (6.6)	109.2, CH_2_	4.69, d (5.1)	20.3, CH_3_	1.03, d (6.5)
11	16.4, CH_3_	0.80, d (6.6)	22.7, CH_3_	1.74, s	21.4, CH_3_	0.86, d (6.5)
12	11.3, CH_2_	2.06, s	59.8, CH_2_	4.06, m	6.3, CH_3_	0.96, d (7.2)
13	55.3, CH	2.43, dd (9.1, 5.4)	64.3, CH	1.64, dd (9.6, 4.9)	54.9, C	1.72, dd (7.9, 5.0)
14a	62.3, CH_2_	3.19, dd (10.5, 9.1)	62.5, CH_2_	3.38, m	62.0, CH_2_	3.85, dd (10.7, 5.0)
14b		3.61, dd (10.5, 5.4)		3.65, dd (10.5, 5.0)		3.50, dd (10.7, 7.9)
15	189.7, CH	9.97, s				

*^a^* Measured in CDCl_3_; *^b^* Measured in methanol-*d*_4_.

**Table 3 jof-08-00009-t003:** ^1^H (600 MHz) and ^13^C (150 MHz) NMR Spectroscopic Data for **7**–**9**.

No.	7 *^a^*		8 *^b^*		9 *^b^*	
	*δ*_C_, Type	*δ*_H_ (*J* in Hz)	*δ*_C_, Type	*δ*_H_ (*J* in Hz)	*δ*_C_, Type	*δ*_H_ (*J* in Hz)
1	137.5, C		137.4, C		57.5, CH	2.54, br s
2	165.3, C		165.3, C		163.7, C	
3	52.6, C		52.5, C		41.8, C	
4a	33.7, CH_2_	1.41, dd (13.4, 5.9)	33.6, CH_2_	1.40, dd (13.3, 5.9)	53.2, CH_2_	1.66, dd (13.2, 10.4)
4b		1.50, dd (13.4, 6.4)		1.48, dd (13.3, 6.5)		2.10, dd (13.2, 10.4)
5a	25.2, CH_2_	0.91, m	25.2, CH_2_	0.90, m	67.0, CH	3.84, m
5b		1.80, m		1.78, m		
6a	44.3, CH	1.06, m	44.2, CH	1.06, m	47.2, CH_2_	1.21, m
6b						1.98, m
7	44.7, CH	3.06, br s	44.5, CH	3.04, br s	32.2, C	
8	19.7, CH_3_	0.97, s	19.6, CH_3_	0.96, s	28.7, CH_3_	0.99, s
9	31.6, CH	1.03, m	31.6, CH	1.03, m	55.0, CH	2.02, br s
10	21.7, CH_3_	1.06, d (5.9)	21.7, CH_3_	1.04, d (5.8)	30.3, CH_3_	1.09, s
11	20.8, CH_3_	0.77, d (5.9)	20.8, CH_3_	0.76, d (5.8)	31.7, CH_3_	0.95, s
12a	11.0, CH_3_	2.06, s	10.9, CH_3_	2.04, s	103.9, CH_2_	4.75, br s
12b						4.97, br s
13	63.6, CH	2.22, d (9.6)	63.4, CH	2.23, d (9.8)	53.2, CH	2.01, br s
14	147.9, CH	6.55, dd (15.9, 9.6)	151.5, CH	6.80, dd (15.4, 9.9)	70.5, CH	4.13, d (6.2)
15	188.1, CH	10.08, s	188.1, CH	10.05, s	74.9, CH	3.59, d (6.2)
16	132.2, CH	6.08, d (15.9)	122.1, CH	5.81, d (15.5)		
17	198.6, C		171.1, C			
18	27.5, CH_3_	2.20, s				

*^a^* Measured in CDCl_3_; *^b^* Measured in methanol-*d*_4_.

**Table 4 jof-08-00009-t004:** ^1^H (600 MHz) and ^13^C (150 MHz) NMR Spectroscopic Data for **10** and **11**.

No.	10 *^a^*		11 *^a^*	
	*δ*_C_, Type	*δ*_H_ (*J* in Hz)	*δ*_C_, Type	*δ*_H_ (*J* in Hz)
1	161.1, C		161.1, C	
2	113.7, CH	6.68, s	113.7, CH	6.69, s
3	147.7, C		147.8, C	
4	108.1, CH	6.75, s	108.1, CH	6.76, s
4a	157.4, C		157.4, C	
5	78.2, CH	5.73, ddd (6.6, 4.4, 3.9)	78.6, CH	5.62, ddd (8.4, 3.8, 1.7)
6a	37.7, CH_2_	3.01, dd (16.2, 4.4)	39.3, CH_2_	3.10, dd (16.3, 8.4)
6b		2.85, dd (16.2, 6.6)		2.99, dd (16.3, 3.8)
7	170.0, C		170.2, C	
8	79.4, CH	5.64, d (3.9)	79.8, CH	5.63, d (1.7)
8a	114.7, C		114.6, C	
9	178.3, C		178.2, C	
9a	109.0, C		109.0, C	
10a	167.7, C		167.4, C	
1′	22.5, CH_3_	2.41, s	22.5, CH_3_	2.42, s
2′	169.5, C		170.1, C	
3′	52.4, CH_3_	3.73, s	52.5, CH_3_	3.78, s
4′	53.0, CH_3_	3.81, s	53.1, CH_3_	3.83, s
1-OH		12.06, s		12.01, s

*^a^* Measured in CDCl_3_; *^b^* Measured in methanol-*d*_4_.

**Table 5 jof-08-00009-t005:** ^1^H (600 MHz) and ^13^C (150 MHz) NMR Spectroscopic Data for **12** and **13**.

No.	12 *^a^*		13 *^a^*	
	*δ*_C_, Type	*δ*_H_ (*J* in Hz)	*δ*_C_, Type	*δ*_H_ (*J* in Hz)
1	221.6, C		221.4, C	
2	50.6, CH	2.16, m	50.6, CH	2.13, m
3	42.1, C		42.0, C	
4a	36.0, CH2	1.45, m	36.1, CH_2_	1.47, m
4b		1.66, m		1.67, m
5a	26.0, CH2	0.84, m	26.0, CH_2_	0.83, m
5b		1.78, m		1.79, m
6	50.2, CH	1.29, m	50.1, CH	1.28, m
7	51.5, CH	2.56, br s	51.5, CH	2.62, br s
8	22.1, CH3	1.08, s	22.1, CH_3_	1.09, s
9	29.9, CH	1.41, m	30.0, CH	1.43, m
10	20.4, CH3	0.77, d (6.6)	20.4, CH_3_	0.78, d (6.7)
11	21.3, CH3	0.89, d (6.4)	21.3, CH_3_	0.92, d (6.5)
12	6.5, CH3	0.95, d (7.2)	6.5, CH_3_	0.96, d (7.2)
13	51.6, CH	1.90, m	51.6, CH	1.94, m
14a	65.3, CH2	4.05, dd (11.3, 5.1)	65.5, CH_2_	4.04, dd (11.3, 5.2)
14b		4.35, dd (11.3, 5.1)		4.37, dd (11.3, 5.2)
1′	161.1, C		161.1, C	
2′	113.6, CH	6.67, s	113.7, CH	6.68, s
3′	147.6, C		147.8, C	
4′	108.1, CH	6.75, s	108.1, CH	6.75, s
4a′	157.3, C		157.4, C	
5′	78.2, CH	5.67, ddd (6.4, 4.3, 3.9)	78.5, CH	5.59, ddd (8.2, 3.9, 1.8)
6′a	37.7, CH2	2.99, dd (16.1, 4.3)	39.2, CH_2_	3.07, dd (16.3, 8.2)
6′b		2.84, dd (16.1, 6.4)		2.99, dd (16.3, 3.9)
7′	169.5, C		170.0, C	
8′	79.5, CH	5.59, d (3.9)	79.9, CH	5.58, d (1.8)
8a′	114.5, C		114.5, C	
9′	178.2, C		178.2, C	
9a′	109.0, C		109.0, C	
10a′	167.7, C		167.3, C	
1″	22.5, CH3	2.40, s	22.5, CH_3_	2.41, s
2″	169.4, C		169.7, C	
3″	52.4, CH3	3.72, s	52.5, CH_3_	3.72, s
1′-OH		12.06, s		12.06, s

*^a^* Measured in CDCl_3_; *^b^* Measured in methanol-*d*_4_.

**Table 6 jof-08-00009-t006:** Inhibitory effects of the isolates against five plant pathogens (MIC, μg/mL) ^a^.

Compd	Psa	*P. infestans*	*A. solani*	*R. solani*	*F. oxysporum*
**3**	256	NA	128	256	NA
**4**	NA ^c^	128	NA	NA	NA
**7**	128	NA	64	128	256
**8**	256	NA	256	NA	NA
**9**	NA	128	NA	NA	NA
**10**	64	128	NA	NA	NA
**11**	128	64	NA	NA	NA
**12**	256	64	NA	64	NA
**13**	128	32	NA	NA	NA
**14**	NA	NA	8	NA	128
**15**	16	NA	16	NA	NA
**16**	128	128	128	256	NA
Streptomycin ^b^	8	−	−	−	−
Hygromycin B ^b^	−	8	4	16	32

^a^ Compounds without any bioactivity are not listed; ^b^ Positive controls; ^c^ NA = no activity at 256 μg/mL.

## Data Availability

X-ray crystallographic data of **1**, **3**, **9**, and **18** (CIF) is available free of charge at https://www.ccdc.cam.ac.uk (accessed on 1 December 2021).

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
