# Peer review of "Sesquiterpenoids and Xanthones from the Kiwifruit-Associated Fungus Bipolaris sp. and Their Anti-Pathogenic Microorganism Activity"

_jof, 2021, doi:10.3390/jof8010009_

Round 1

Reviewer 1 Report

In this manuscript, 18 compounds including 13 new ones were characterized from kiwi 476 endophytic fungus Bipolaris sorokiniana, as well as their anti-Psa activity and antifungal activity against plant pathogens were evaluated. I would like to recommend the acceptance of the manuscript after a minor revision.

  1. The authors should revise the tense (past simple) 
  2. Page 2, Line 42-47: the authors should rephrase these sentences as they are mentioning their previous report/investigation (e.g. Based on this strategy, we previously…)
  3. Line 49: revise “currently” e.g. In the current investigation, we…
  4. Page 3, Line 89: the plant name should be italic
  5. I suggest that the MIC in the manuscript be expressed in micromole (µM)
  6. Page 10, Line 337: revise the sentence

Author Response

Dear the Reviewer,

Thank you for the comments on our manuscript. We have revised the manuscript carefully according to all comments. Details for response to comments are attached below. In addition, the English has been polished, while some minor errors are also corrected in the revised manuscript.

Sincerely,

Juan He

1. The authors should revise the tense (past simple) 

Re. We have polished the English throughout the whole text.

2. Page 2, Line 42-47: the authors should rephrase these sentences as they are mentioning their previous report/investigation (e.g. Based on this strategy, we previously…) 

Re. We have revised it.

3. Line 49: revise “currently” e.g. In the current investigation, we…

Re. It has been revised.

4. Page 3, Line 89: the plant name should be italic

Re. It has been corrected.

5. I suggest that the MIC in the manuscript be expressed in micromole (µM)

Re. To our knowledge, MIC is suitable for antibacterial or antifungal assay, while µM is used for cytotoxicity or other enzyme activity assays.

6. Page 10, Line 337: revise the sentence

Re. It has been corrected.

Reviewer 2 Report

This paper describes several interesting new compounds from a plant-associated fungus that was apparently obtained as an endophyte from kiwifruit in China. The description of the isolation and structure elucudation ae well written and appear sound to me, even though I have not checked these parts in detail. I assume that specialists for structure elucidation will be invited in parallel for review.

However, I have added some comments on the biological aspects in the attached pdf because the "identification" of the fungus sucks.

The authors could avoid trouble if they just call the fungus "Bipolaris sp."

It is not necessary that they include a complete species description but the authors must make sure that their strain is available to specialists who can provide an accurate identification later on. .

This is a mycological journal and it should not happen that the taxonomy does not meet the standards. The authors are invited to consult these articles:

https://pubmed.ncbi.nlm.nih.gov/34273757/

https://pubmed.ncbi.nlm.nih.gov/28199101/

In addition, I have also commented on some other aspects that need to be clarified. Oomycetes are not fungi etc.

Author Response

Dear the Reviewer,

We do appreciate that you could give kind comments on our manuscript, especially the professional advice on the identification of fungal species is very worthy of our study. This makes us realize that even if we are engaged in natural product research, we must be very cautious about resource traceability, we will pay much attention to our work in future.

According to the suggestions (as well as those given in the attached PDF file), we have made a comprehensive modification to manuscript, changing the fungus to Bipolaris sp., and changing the "Kiwi endophytic fungus" to "kiwi-associated fungus" in the text. In addition, sections for “chemical investigation” have been checked carefully according to the comments from the Reviewer 1, while the English has been polished.

Sincerely,

Juan He

Round 2

Reviewer 2 Report

The paper has now improved and I would recommend acceptance. The authors should, however, be careful because they have introduced some typos in the revised version. For instance the plural of microorganism is "microorganisms" which has been written wrong in two instances. But I think that this can be taken care of during the copy editing process.